# The Development of Peritoneal Metastasis from Gastric Cancer and Rationale of Treatment According to the Mechanism

**DOI:** 10.3390/jcm11020458

**Published:** 2022-01-17

**Authors:** Yutaka Yonemura, Haruaki Ishibashi, Akiyoshi Mizumoto, Gorou Tukiyama, Yang Liu, Satoshi Wakama, Shouzou Sako, Nobuyuki Takao, Toshiyuki Kitai, Kanji Katayama, Yasuyuki Kamada, Keizou Taniguchi, Daisuke Fujimoto, Yoshio Endou, Masahiro Miura

**Affiliations:** 1NPO to Support Peritoneal Surface Malignancy Treatment, Asian School of Peritoneal Surface Malignancy Treatment, 510, Fukushima-Cho, Kyoto 600-8189, Japan; 2Department of Regional Cancer Therapy, Peritoneal Dissemination Center, Kishiwada Tokusyukai Hospital, Kishiwada 596-8522, Japan; ishibashi@kusatsu-gh.or.jp (H.I.); gorotukiyakuza.007@gmail.com (G.T.); lymikeleo@hotmail.com (Y.L.); akebono_jaguar@yahoo.co.jp (S.W.); s-sako.rgb-sh009@docomo.ne.jp (S.S.); kitait@kcn.jp (T.K.); kanji@u-fukui.ac.jp (K.K.); y_kamada@kuhp.kyoto-u.ac.jp (Y.K.); 3Department of Regional Cancer Therapy, Peritoneal Dissemination Center, Kusatsu General Hospital, Kusatsu 525-8585, Japan; mizumotoakiyoshi1206@yahoo.co.jp (A.M.); nt421500@gmail.com (N.T.); 4Department of Surgery, Mizonokuchi Hospital, Teikyo University School of Medicine, Kawasaki 213-8570, Japan; keizotng16@me.com (K.T.); rui5218@gmail.com (D.F.); 5Central Research Resource Center, Cancer Research Institute, Kanazawa 922-1192, Japan; yendo2@staff.kanazawa-u.ac.jp; 6Department of Anatomy, Oita Medical University, Kasama-Machi, Oita 879-5593, Japan; miura@oita-u.ac.jp

**Keywords:** gastric cancer, peritoneal metastasis, peritoneal dissemination, intraperitoneal chemotherapy

## Abstract

In the present article, we describe the normal structure of the peritoneum and review the mechanisms of peritoneal metastasis (PM) from gastric cancer (GC). The structure of the peritoneum was studied by a double-enzyme staining method using alkaline-phosphatase and 5′-nucreotidase, scanning electron microscopy, and immunohistological methods. The fundamental structure consists of three layers, mesothelial cells and a basement membrane (layer 1), macula cribriformis (MC) (layer 2), and submesothelial connective tissue containing blood vessels and initial lymphatic vessels, attached to holes in the MC (layer 3). Macro molecules and macrophages migrate from mesothelial stomata to the initial lymphatic vessels through holes in the MC. These structures are characteristically found in the diaphragm, omentum, paracolic gutter, pelvic peritoneum, and falciform ligament. The first step of PM is spillage of cancer cells (peritoneal free cancer cells; PFCCs) into the peritoneal cavity from the serosal surface of the primary tumor or cancer cell contamination from lymphatic and blood vessels torn during surgical procedures. After PFCCs adhere to the peritoneal surface, PMs form by three processes, i.e., (1) trans-mesothelial metastasis, (2) trans-lymphatic metastasis, and (3) superficial growing metastasis. Because the intraperitoneal (IP) dose intensity is significantly higher when generated by IP chemotherapy than by systemic chemotherapy, IP chemotherapy has a great role in the treatment of PFCCs, superficial growing metastasis, trans-lymphatic metastasis and in the early stages of trans-mesothelial metastasis. However, an established trans-mesothelial metastasis has its own interstitial tissue and vasculature which generate high interstitial pressure. Accordingly, it is reasonable to treat established trans-mesothelial metastasis by bidirectional chemotherapy from both IP and systemic chemotherapy.

## 1. Introduction

Gastric cancer (GC) is the second leading cause of death from cancer. Peritoneal metastasis (PM) is the most common form of metastasis in GC, and PM is about 14% of primary GC cases [1,2]. However, patients have a median survival time of 3–6 months [1,2]. Until the early 1990s, GC with PM was considered an incurable disease because it could not be cured by surgery or systemic chemotherapy alone [3,4]. Even after complete resection of a small number of PMs with gastrectomy plus lymph adenectomy, residual micrometastasis on the peritoneal surface always proliferates and almost all patients will die. Systemic chemotherapy using modern drugs has limited effects on PM [3,4] because only small amounts of systemically administered drugs can enter the peritoneal cavity, and even effective regimens are inevitably interrupted due to the development of side effects or regrowth of multidrug-resistant cancer cells.

In the late 1990s, a combination of cytoreductive surgery (CRS) and perioperative chemotherapy for the treatment of PM was proposed as a comprehensive treatment by the Peritoneal Surface Oncology Group International (PSOGI) [5]. In CRS, all the macroscopically detectable tumors are removed by D2 gastrectomy and peritonectomy [5,6]. However, even macroscopic complete cytoreduction leaves invisible micrometastases in most cases [7]. To eradicate the micrometastases before and after CRS, neoadjuvant IP chemotherapy and intraoperative hyperthermic intraperitoneal chemoperfusion (HIPEC) were developed [8,9,10,11]. The comprehensive treatment improved the long-term survival of GC patients with PM and 10% of patients survived longer than 10 years [11]. Accordingly, the treatment is now considered a curative approach [6,9,11]. For the development of more effective treatments to improve survival, it is important to clarify the mechanisms of the formation of PM.

The present chapter presents the mechanisms of PM from GC, and the rationale for eliminating micrometastasis by chemotherapy.

## 2. Normal Structure of Peritoneum

Macroscopically normal peritoneal parts were obtained from the resected specimens of GC patients and we studied the structures of the lymphatic vascular system by immunohistochemistry and the double enzyme staining method using alkaline-phosphatase (ALP) and 5′-nucleotidase (Nase) reactions. The study was acknowledged by the ethical committee of Kishiwada Tokusyukai Hospital with the study number 19–35, entitling us to a clinical study of the efficacies of a comprehensive treatment of peritoneal metastasis.

The peritoneal cavity normally has 40 to 50 mL of ascites that always circulate through the submesothelial lymphatic vascular system, also known as the peritoneal systemic circulation. Ascites and the glycocalyx such as hyaluronic acid produced from mesothelial cells play a role as a lubricant for the viscera to slide over the visceral and parietal peritoneal surface without friction. Ascites play a role as a medium through which peritoneal macrophages pass to patrol the peritoneal microenvironment [12,13,14].

Figure 1 shows the fundamental structure of the peritoneum. Mesothelial cells cover the basement membrane (BM), which in turn covers a layer of the collagen plate known as the macula cribriformis (MC). Lymphatic vessels permeate the shallow subperitoneal space in Morrison’s pouch (Figure 2A–C and Figure 3), the falciform ligament (Figure 4A–C), the pelvic peritoneum (Figure 5) and the para-colic gutter (Figure 6) [12]. The blood–peritoneal barrier (BPB) consists of mesothelial cells, BM, MC, and connective tissues between mesothelial cells and submesothelial blood vessels (Figure 1). Some submesothelial lymphatic vessels usually look like a blind loop (Figure 4), and are attached to the small holes in the MC and mesothelial cell gaps. These mesothelial cell gaps are named lymphatic stomata (Figure 1, right) [14]. Peritoneal fluid-containing electrolytes are absorbed from the subperitoneal lymphatic vascular system, and macromolecules and inflammatory cells are resorbed and migrate through lymphatic stomata [14,15]. Alkaline-phosphatase and 5′-nucleotidase (5′-Nase) double enzyme staining differentiates subperitoneal blood capillaries (Figure 2A, blue) from lymphatic vessels (brown) (Figure 2A). Below the submesothelial BM, multiple holes (Figure 7) are found in the MC, and the blind loops of the submesothelial lymphatic vessels are attached to the holes (Figure 1, Figure 2, Figure 3, Figure 4, Figure 5, Figure 6, Figure 7 and Figure 8). Activated carbon (CH44) is present in the blind loop of the submesothelial lymphatic vessels (also known as initial lymphatic vessels, which are stained by 5′-Nase enzyme staining (Figure 8)) 2 days after IP injection (Figure 8, left). CH44 is absorbed between the lymphatic endothelial cells (Figure 8, right) and is in the subperitoneal lymphatic vessels in Morrison’s pouch (5′-Nase enzyme staining, Figure 3). These results indicate that intraperitoneally injected CH44 is absorbed through initial lymphatic vessels [14]. These lymphatic vascular structures are found in Morrison’s pouch (Figure 3 and Figure 7), the pelvic peritoneum (Figure 5) and falciform ligament (Figure 4) and have a three-layered fundamental anatomical structure (Figure 6) that includes a first layer consisting of mesothelial cells and BM, a second layer known as the MC, and a third layer containing the initial lymphatic vessels that are attached to holes in the MC. Macro molecules and macrophages migrate from mesothelial stomata and move into the initial lymphatic vessels through holes in the MC (Figure 6).

The lymphatic vascular system of the diaphragm is slightly different from that of the pelvis and Morrison’s pouch. On the abdominal surface of the diaphragm, the openings in the mesothelium that connect it with the submesothelial lymphatic vessels (initial lymphatic vessels) are defined as lymphatic stomata, whereas unattached mesothelial gaps to the initial lymphatic vessels are not defined as lymphatic stomata (Figure 9C). Negative pressure due to inspiration helps to absorb peritoneal fluid and macro molecules through the lymphatic stomata of the diaphragm, and the absorbed materials migrate in the lymphatic vessels, which vertically run through diaphragmatic muscle (Figure 9A). Blood vessels below the mesothelial cells have a role in the absorption of peritoneal fluid (Figure 9B). As shown in Figure 10, holes in the diaphragmatic MC connect with the initial lymphatic vessels (Figure 10).

The greater and lesser omentum have specialized structures know as omental milky spots (OMS), which are associated with inflammatory cell migration and ascite absorption. The OMS are small organs 15 to 800 m in diameter, and their mean number is 35/cm^2^ in infants and 2/cm^2^ in adults [15,17]. The outer surface of OMS is covered with cuboidal mesothelial cells punctured by stomata and supported by a BM (Figure 11 and Figure 12), and mesothelial cells surrounding OMS are flat (Figure 12, left). The MC, glomerular blood vessels and initial lymphatic vessels are found in the second and third layers (Figure 11). The fundamental structure of OMS is similar to that of diaphragm/falciform ligaments and the peritoneum of the pelvis, para-colic gutter and Morrison’s pouch (Figure 6). The number of OMS change with age and peritoneal inflammatory status (Figure 11) [17].

## 3. Mechanisms of the Formation of PM from GC

PM is characterized by a multi-step process consisting of (1) detachment of cancer cells from the serosal surface of the primary GC, (2) migration of peritoneal free cancer cells (PFCCs) through ascites on the distant peritoneal surface and attachment of PFCCs on the peritoneal surface by adhesion molecules, (3) proliferation of PFCCs on the peritoneal surface or invasion of PFCCs into the subperitoneal tissue through the concerted actions of matrix-digesting enzymes, adhesion molecules and motility factors expressed from cancer cells and stromal cells, and (4) neoplastic proliferation accompanying immature stromal elements and angiogenesis [18].

In 1997, Sugarbaker reported two patterns in PM formation concepts, i.e., randomly proximal distribution and redistribution pattern [19].

Yonemura, Y. et al. reported three new PM formation concepts, i.e., (1) trans-mesothelial metastasis, (2) trans-lymphatic metastasis, and (3) superficial growing metastasis [14].

### 3.1. Mechanisms of the Cell Spillage into the Peritoneal Cavity

Spillage of cancer cells into the peritoneal cavity occurs from the serosal surface of the primary tumor or cancer cell contamination from lymphatic and blood vessels torn during surgical procedures.

In the process of cancer cell detachment from the primary tumor, dysfunction of homophilic cell–cell adhesion molecules plays an important role. Analyses of tight and adherence junction molecule expression demonstrate that a reduction in the expression of claudin, occludin and E-cadherin induces cell dispersion [20,21,22,23] and is frequently found in poorly differentiated adenocarcinomas of the stomach [24,25,26]. In GC, poorly differentiated adenocarcinomas have a significantly higher PM potential than the differentiated type and are characterized by downregulation of E-cadherin and its catenin partner [25,26,27,28]. E-cadherin expressed on the adherence junction plays a major role in cell–cell adhesion. The loss of cadherin induces the loosening of the adhesion, resulting in the dispersion of cells.

### 3.2. Adhesion of PFCCs to the Distant Peritoneum

PFCCs migrate through ascites and reach the distant peritoneum (Figure 13). While rolling on mesothelial cells, PFCCs adhere to mesothelial cells via adhesion molecules and their ligands. In homophilic heterotypic adhesion, P-cadherin participates in the loose attachment of PFCCs to mesothelial cells [29].

Integrin L2 and 2 expressed on PFCCs heterotopically bind with PCAM-1 and VCAM-1 from mesothelial cells [14,30].

Hyaluronic acids from mesothelial cells are specific ligands of the transmembrane glycoprotein CD44 from PFCCs, and CD44 isoforms enhance local growth and metastasis of cancer cells [31,32].

The Syalyl Lewis A antigen has the carbohydrate structure on cancer cells needed for E-selectinbinding to mesotherial cells [33].

CA 125 is a mucin-like glycoprotein that is upregulated in some cancer cells [34,35], whereas mesothelin is expressed by normal mesothelial cells and binds with CA125 [36].

Recently, Arita T et al. reported that tumor-derived exosomes (TEX) may play a critical role in the development of PM, which may be due to inducing increased expression of adhesion molecules from mesothelial cells [37]. TEX are known to contain mRNA and micro RNA from GC cells, and these may trigger mesothelial cell transformation, leading to a cancer-preferable microenvironment.

Mesothelial cells are flat and squamous in shape and are maintained in close contact by tight junctions (Figure 14A). When PFCCs appear in the peritoneal cavity, mesothelial cells are activated by PFCC-produced cytokines (Figure 14B–D) [38,39]. IL-6, IL-1, granulocyte-colony stimulating factor (G-CSF), IL-15, IL-1, INF-, and epidermal growth factor (EGF) all reportedly induce changes in mesothelial cell morphology. These cytokines are produced from not only cancer cells but also host inflammatory cells and fibroblasts [40,41,42,43].

When the submesothelial BM is exposed by mesothelial cell shrinkage, PFCCs attach to the BM by adhering to integrin molecules expressed on the surface of microvilli (Figure 15A,B).

BM consists of laminin, type IV collagen, heparin sulfate, proteoglycan, entactin, and perlecan. VLA-2, and -3 have roles in PFCC adhesion to basement membrane components [44,45,46,47] and can bind to collagen, laminin, and fibronectin [48].

After PFCC attachment to the BM, PM can be established by three processes: (1) trans-mesothelial metastasis, (2) trans-lymphatic metastasis, and (3) superficial growing metastasis [14].

### 3.3. Mechanisms of Trans-Mesothelial Metastasis

After PFCCs’ attachment to mesothelial cells, PFCCs digest the BM exposed between shrunk mesothelial cells. Akedo H et al. observed three growth patterns of cancer cells when rat hepatoma cell monolayers were co-cultured with a rat mesothelial cell monolayer [41]. Tumor cells either formed “pile-up” nests on the mesothelial monolayer, exhibited invasive growth between adjacent mesothelial cells, or failed to attach and grew in suspension.

Figure 16 shows a micrometastasis of a differentiated adenocarcinoma growing on the small bowel mesentery. Cancer cells show Ki 67 immunoreaction, suggesting high proliferative activity (Figure 16B). CD34-positive stromal cells are observed at the invasion front of the micrometastasis (Figure 16C). CD34 is a transmembrane glycoprotein that is expressed on the stem cells and fibroblasts of immature tissues [49]. As shown in Figure 16D, no newly formed blood vessels are detected in the stroma of micrometastasis.

In contrast, Figure 17 shows the histological structure of an established metastasis from a differentiated adenocarcinoma. The cancer cells show high proliferative activity (Figure 17B), and the stromal cells are CD34-positive stromal cells (Figure 17C), suggesting the immaturity of the interstitial tissue. Newly formed blood vessels are found in the stroma of established PM (Figure 17D). These results indicate that PFCCs attach to the peritoneal surface and invade the subperitoneal tissue by a mechanism involving the concerted expression of motility factors [50,51,52,53,54,55,56] matrix digesting enzymes [57,58,59,60,61,62,63,64,65,66,67,68,69], and adhesion molecules [44,45,46,70,71]. As PFCCs grow in the subperitoneal tissue, a new vascular network forms in response to the secretion of angiogenesis factors from cancer cells and surrounding interstitial cells (Figure 17) [72,73,74]. As shown in Figure 17C,D, CD34-positive interstitial fibroblasts are found in the stromal tissue, and newly formed blood vessels are detected in the stromal tissues of established trans-mesothelial metastasis.

Figure 18 shows the micrometastasis of a poorly differentiated adenocarcinoma that formed in the rectal Douglas pouch. Cancer cells invading from the peritoneal surface are found in the subserosal layer and invade into the rectal muscle layer (Figure 18A) and submucosal layer (Figure 18B). These cancer cells are immunoreactive with Ki67 monoclonal antibody, suggesting high proliferative activity (Figure 18C). CD34-positive cells are detected in the stromal tissue (Figure 18D), indicating that the poorly differentiated adenocarcinoma easily invades through the gaps in the muscle layer by a mechanism that uses motility factors, adhesion molecules and matrix digesting enzymes. Metastasis will become established upon cancer cell proliferation with angiogenesis induced by CD34-positive stromal cells.

This type of peritoneal metastasis is named trans-mesothelial metastasis.

### 3.4. Mechanisms of Superficial Growing Metastasis

Cancer cells located greater than 100 m from blood vessels will die due to an insufficient supply of oxygen [18,42,75]. The BPB occupies the space between the mesothelial cell surface and submesothelial blood vessels (Figure 1). In Morrison’s pouch, the falciform ligament, and diaphragmatic peritoneum, there are subperitoneal vascular networks (a short-distance BPB), located just beneath the MC (Figure 2 and Figure 9) [12] and PFCCs with low invasive activity proliferate by absorbing oxygen from the subperitoneal blood vessels on the peritoneal surface with a short-distance BPB [12]. This type of metastasis is named the superficial growing metastasis (Figure 19) [14].

In contrast, the peritoneum on the anterior abdominal wall (Figure 20A) has few submesothelial blood vessels (Figure 20C). The peritoneum covering the anterior abdominal wall is considered to be resistant to the development of superficial growing metastasis because of its long-distance BPB (Figure 20).

### 3.5. Mechnisms of Trans-Lymphatic Metastasis

In trans-lymphatic metastasis, PFCCs migrate into the initial lymphatic vessels. Cytokine secretion from the PFCCs causes mesothelial cell shrinkage (Figure 14) and hole formation in the MC (Figure 21A), thereby exposing the initial lymphatic vessel (Figure 6, Figure 7, Figure 21 and Figure 22), and resulting in a direct route from the peritoneal cavity to the initial lymphatic vessels (Figure 6 and Figure 22A). Scanning electron micrography shows that the PFCCs migrate into the initial lymphatic vessels through mesothelial lymphatic stoma and holes in the MC (Figure 21A–C).

Schematic diagram of trans-lymphatic metastasis is shown in Figure 22A. PFCCs invade the initial lymphatic vessels and proliferate in the lymphatic lacunae (Figure 22B), and their growth finally destroys the lymphatic vessels (Figure 22C). Trans-lymphatic metastasis develops in the initial lymphatic vessel-rich sectors in the parietal peritoneum, showing a pink area (Figure 23). As shown in Figure 20B, initial lymphatic vessels are not found in anterior abdominal wall, and trans-lymphatic metastasis is not found in the sector.

OMS are common sites of trans-lymphatic metastasis [12,13,14,15,76]. PFCCs are adsorbed to the stomata between the cuboidal mesothelial cells covering the OMS and invade the omental lymphatic vessels through holes in the MC (Figure 11 and Figure 12).

On the small bowel mesentery, 2–3 cm in from the attachment sites on the small bowel, milky spot-like structures are found (Figure 24). PM from GC is frequently found in the peritoneal area (Figure 24A), and intraperitoneally injected CH44 is adsorbed in the same area of the small bowel mesentery (Figure 24B), suggesting the absorption of CH44 by the initial lymphatic vessels in the milky spots of small bowel mesentery. Scanning electron micrography shows the oval-shaped structure of the milky spot-like structure covered by cuboidal mesothelial cells (Figure 24C). Below the cuboidal mesothelial cells, MC with holes is detected (Figure 24D).

Scanning electron micrography shows the omental milky spots (oval-shaped structures) covered with cuboidal mesothelial cells. Between the cuboidal mesothelial cells, stomata-like holes are observed. Initial lymphatic vessels can be seen in the epiploic appendage of the colon (Figure 25A), and metastasis on the epiploic appendage should be removed (Figure 25B,C).

### 3.6. Pocket-Like Structures in the Peritoneal Cavity, Where PFCCs Are Trapped

The Douglas pouch and rectovesical pouch are well-known metastasis sites of GC because PFCCs are held for a long time on the peritoneal surface by gravity, resulting in the establishment of PM. The peritoneal cavity has several pocket-like structures, where PFCCs are trapped. PFCCs migrate into the pocket-like structures via peritoneal fluid. During their stay in the pockets, PFCCs establish an environment conducive to trans-mesothelial metastasis by causing mesothelial cell separation.

Figure 26 shows the omental bursa consisting of the omental sac (Figure 26A,B), superior recess (Figure 26C), and anterior vestibule (Figure 26D). PFCCs migrate through the foramen of Winslow into the omental bursa (Figure 26).

Around the duodenojejunal junction, there are two pockets, i.e., the superior and inferior recess (Figure 27). Figure 27C shows a metastasis on the inferior recess.

In the left side of sigmoid mesocolon, there is an intersigmoid recess (Figure 28). The number of recesses varies from case to case (Figure 28B,C). Figure 28 C shows the metastases on the intersigmoid recess.

The pelvic cavity has many pockets. Males have a rectovesical pouch (Figure 29A), and females have the Douglas pouch and vesicouterine pouch (Figure 29B). Figure 29C is in intraperitoneal view of the en-bloc resection pelvic pockets, and Figure 29D shows the resected specimen of the rectum, uterus and pelvic peritoneum.

## 4. Selection of the Route of Chemotherapy with Special Reference to the Mechanisms of PM formation

Systemic chemotherapy has little effect on PM because the intraperitoneal (IP) transport of chemotherapeutic agents is limited [77]. BPB with average distance of 90 μm hinders the movement of drugs from the subperitoneal blood vessels to the peritoneal cavity [77]. In contrast, IP chemotherapy could generate significantly higher intraperitoneal dose intensity than systemic chemotherapy (Table 1) [78]. After IP administration, the drug penetration distances into the subperitoneal tissue are different from drug to drug (Table 1). In general, the larger the molecular weight of the drug, the longer the drug stays in the peritoneal cavity. Drugs with higher molecular weights are recommended for IP chemotherapy to treat PFCCs, superficial growing metastasis (Figure 19), trans-lymphatic metastasis (Figure 22) and the early stage of trans-mesothelial metastasis (Figure 16), where the cancer cell is growing in the superficial submesothelial layer without angiogenesis (Figure 16).

Taxans show anti-cancer effects to injure the function of the spindle body by promoting microtubule polymerization. Because taxol and docetaxel have hydrophobic properties, these drugs are conjugated with cremophol EL and polysorbate 80 to be water-soluble components [78]. Taxol and docetaxel are gradually released from the carriers to ascites after intraperitoneal administration. Accordingly, these drugs tend to stay for a long time after intraperitoneal administration. The concentrations of these drugs in ascites are maintained for 24 h at significantly high levels to kill cancer cells [78]. Accordingly, these drugs may be effective in the treatment of PFCCs, superficial growing metastasis, and trans-lymphatic metastasis (Table 1) [78,79]. For the treatment of trans-mesothelial metastasis, intraperitoneal administration of drugs capable of deeper penetration into the subperitoneal tissue is recommended (Table 1). After IP administration, cisplatin, oxaliplatin, and carboplatin can penetrate a depth greater than 1 mm [80,81]. However, an established trans- mesothelial metastasis has its own interstitial tissue and vasculature, resulting in high interstitial pressure in the metastasis (Figure 17 and Figure 18) [42]. Accordingly, drugs that penetrate a metastasis with high interstitial pressure are easily excreted into the normal tissue with low interstitial pressure. A bidirectional chemotherapy from both sides of the intraperitoneal and systemic administration is a reasonable option for treating established trans-mesothelial metastasis (Figure 17) [7,8,10]. Metastases invading the submucosal layer and proper muscle of bowel should be treated with systemic chemotherapy (Figure 18).

Because chemotherapeutic drugs can injure proliferating cancer cells but penetrate the peritoneal surface only a short distance, IP chemotherapy should be repeated until drug resistance appears or macroscopic complete cytoreduction can be performed.

## 5. Effects of Neoadjuvant Intraperitoneal and Systemic Chemotherapy (NIPS) on Lymph Node Metastasis

IP chemotherapy is considered an effective method of treating PM from GC [7,8]. Yonemura Y et al. developed a novel neoadjuvant chemotherapy combining intraperitoneal and systemic chemotherapy, which is named NIPS [7,8,11]. In NIPS, intraperitoneal administrations of 40 mg of docetaxel and cisplatinum with 500 mL of normal saline are performed on day 1 and 14, and oral intake of 60 mg/m^2^ of S1 starts from day 1 to day 14. After 1 week rest (from day 15 to day 21), NIPS repeats for at least 3 cycles. One month after the last cycle of NIPS, patients opted for performing cytoreductive surgery and received a gastrectomy plus D2 lymph adenectomy and peritonectomy. They studied the effects of lymph node metastasis (LNM) after NIPS and compared the results to the no NIPS group [82].

The incidence of N0 cases was significantly higher in the NIPS group (37/107; 34.6% vs. 14/136; 10.3%) (*p* < 0.0001). Survival was significantly longer after NIPS plus cytoreductive surgery than in the non-NIPS group. NIPS is a very effective method of controlling LNM from GC. After IP administration of chemotherapeutic drugs, extremely higher concentrations of chemotherapeutic drug are absorbed through OMS and the efferent lymphatic fluid drains into the regional lymph nodes of the stomach. As a result, LNM from GC is exposed to much higher concentrations of chemotherapeutic drugs than when chemotherapy is given systemically. This feature of the lymphatic circulation accounts for the much greater effects of NIPS on LNM.

## 6. Effects of NIPS on PM from GC

Effects of NIPS on PM from GC has been reported from Japanese surgical oncologists. Ishigami et al. conducted a randomized phase III trial to confirm the effects of NIPS. Patients were randomly assigned to two groups, i.e., (1) IP and intravenous (IV) paclitaxel plus S-1 (IP; IP paclitaxel 20 mg/m^2^ and IV paclitaxel 50 mg/m^2^ on days 1 and 8 plus S-1 80 mg/m^2^ per day on days 1 to 14 for a 3-week cycle) and (2) S-1 plus IV cisplatin (SP; S-1 80 mg/m^2^ per day on days 1 to 21 plus cisplatin 60 mg/m^2^ on day 8 for a 5-week cycle).

This trial failed to show statistical superiority of survival in the NIPS group as compared with the control group. However, the response rate analyses suggested possible clinical benefits of NIPS for PM from GC [83].

Yonemura Y et al. studied the effects of NIPS by laparoscopy, and the peritoneal cancer index (PCI) was significantly reduced after three cycles of NIPS. These results indicate that NIPS is effective in PCI reduction [84]. The survival of GC patients with PM treated by NIPS plus CRS was significantly better than those treated with cytoreduction without NIPS [12]. Additionally, Yonemura et al. reported that post-NIPS PCI ≤6 was the strongest independent prognostic factor [10,11]. Accordingly, NIPS is essential to improve postoperative survival of GC patients with PM by reducing PCI and micrometastasis left on the preserved peritoneal surface after CRS.

## 7. Effects of HIPEC on PM from GC

Yonemura et al. studied the effect of laparoscopic HIPEC (LHIPEC) on PM from GC. LHIPEC receiving cisplatin 50 mg and docetaxel 40 mg in 4000 mL of normal saline at 43 ± 0.5 °C for 60 min were performed in 55 GC patients with PM, and laparoscopy was again performed one month later. PCI was significantly reduced and the positive cytology became negative in 62% of patients with positive cytology at the time of LHIPEC [84].

Brandl A collected and analyzed a worldwide cohort of patients treated with cytoreductive surgery and HIPEC with long-term survival in order to explore relevant patient characteristics [85]. From an analysis of 448 patients, a total of 28 patients with a mean PCI of 3.3 survived longer than 5 years after CRS plus HIPEC. The overall median survival was 11.0 years (min 5.0; max 27.9). The predictor completeness of cytoreduction (CC-0) and PCI ≤ 6 were present in 22/28 patients. They concluded that the completeness of cytoreduction and low PCI seemed to be crucial and that long-term survival and even cures are possible in patients with PM of GC treated with CRS and HIPEC

Yang XJ performed a randomized phase-III study to evaluate the efficacy and safety of CRS plus HIPEC for PM from GC [86]. Sixty-eight gastric PC patients were randomized into CRS alone (*n* = 34) or CRS plus HIPEC (*n* = 34) receiving cisplatin 120 mg and mitomycin C 30 mg each in 6000 mL of normal saline at 43 ± 0.5 °C for 60–90 min. The median survival was 6.5 months in CRS and 11.0 months in the CRS plus HIPEC groups (*p* = 0.046). Multivariate analysis found that CRS plus HIPEC, synchronous PC, CC 0–1, systemic chemotherapy ≥ 6 cycles, and no serious adverse events were independent predictors for better survival. They concluded that CRS + HIPEC with mitomycin C 30 mg and cisplatin 120 mg may improve survival with acceptable morbidity [86].

These results indicate that HIPEC is directly effective on PM from GC, and HIPEC may improve GC patients’ survival after complete resection of the PM.

## 8. Future Perspectives

IP immunotherapy offers a novel approach for the control of regional disease of the peritoneal cavity by breaking immune tolerance. These strategies include heightening T- cells and vaccine induction of anti-cancer memory against tumor-associated antigens [87].

Catumaxomab, a non-humanized chimeric antibody, is characterized by its unique ability to bind to three different types of cells: tumor cells expressing the epithelial cell adhesion molecule (EpCAM), T lymphocytes (CD3) and also accessory cells (Fcy receptor). Because up to 90% of gastric cancers express EpCAM, IP infusion of catumaxomab after complete resection of all macroscopic disease could therefore efficiently treat macroscopic residual disease [88]. Knödler M performed a prospective randomized phase-II study investigating the efficacy of catumaxomab followed by chemotherapy (arm A, 5- fluorouracil, leucovorin, oxaliplatin, docetaxel, FLOT) or FLOT alone (arm B) in patients with GC and PM. However, no survival benefit was found in the treatment group [88].

Nivolumab, an anti-programmed cell death-1 (PD-1) antibody, has been developed and survival benefit was obtained in GC patients with PM [89]. The combination of CRS plus HIPEC and postoperative nivolumab may improve postoperative survival, but no randomized phase-III study was reported.

Nab-paclitaxe (nanoparticle alubumin-bound paclitaxel) has effective transferability to tumor tissues and strong antitumor effects for peritoneal metastasis [90]. Ishikawa M reported that nanoparticle alubumin-bound (nab)-PTX treatment has beneficial effects on the survival of GC patients with PM as compared the RAM plus solvent-based (sb) paclitaxel [90].

Pressurized intraperitoneal aerosol chemotherapy (PIPAC) was introduced as a new treatment for patients with peritoneal metastases in 2011. Adverse events greater than grade 2 occurred after 12–15% of procedures. An objective clinical response of 50–91% was found for gastric cancer (median survival of 8–15 months). Alyami M et al. reported that PIPAC was safe, and the objective response and quality of life were encouraging. Therefore, PIPAC can be considered as a treatment option for refractory, isolated peritoneal metastasis [91]. However, no long-term survival after PIPAC has been reported.

Patients with PM may be improved by gene therapy [92]. Many ideas have been reported, but large-scale clinical studies have not yet been performed.

We are awaiting the development of new effective nanomolecules, cancer-specific antibodies or anti-cancer drugs for the combination of treatment options with CRS.

In the surgical field, further studies are needed in order to improve existing selection criteria for cytoreductive surgery.

## 9. Conclusions

The fundamental structure of the peritoneum consists of three layers: mesothelial cells and the basement membrane (layer 1), the macula cribriformis (MC) (layer 2), and the submesothelial connective tissue containing blood vessels and initial lymphatic vessels (layer 3). Macro molecules and macrophages migrate from mesothelial stomata to the initial lymphatic vessels through holes in the MC.

These structures are characteristically found in the diaphragm, omentum, paracolic gutter, pelvic peritoneum, and falciform ligament.

The first step of PM is the spillage of cancer cells (peritoneal free cancer cells; PFCCs) into the peritoneal cavity from the serosal surface of the primary tumor or cancer cell contamination from lymphatic and blood vessels torn during surgical procedures.

After PFCCs adhere to the peritoneal surface, PMs form by three processes, i.e., (1) trans-mesothelial metastasis, (2) trans-lymphatic metastasis, and (3) superficial growing metastasis. Because the intraperitoneal (IP) dose intensity is significantly higher when generated by IP chemotherapy than by systemic chemotherapy, IP chemotherapy plays a great role in the treatment of PFCCs, superficial growing metastasis, trans-lymphatic metastasis and early stage of trans-mesothelial metastasis. However, an established trans-mesothelial metastasis has its own interstitial tissue and vasculature, which generates high interstitial pressure.

Accordingly, it is reasonable to treat established trans-mesothelial metastasis by bidirectional chemotherapy from both IP and systemic chemotherapy.

## Figures and Tables

**Figure 1 jcm-11-00458-f001:**
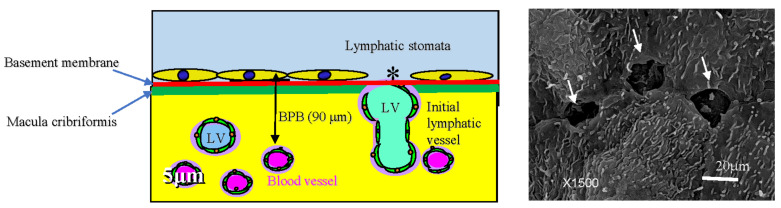
Normal fundamental structure of the peritoneum. Mesothelial cells cover the basement membrane (BM, brown line), and collagen plate named the macula cribriformis (MC, green line) lies just below the basement membrane. Lymphatic vessels (LV) permeate the shallow subperitoneal space (left). The blood–peritoneal barrier (BPB) consists of mesothelial cells, BM, MC, and connective tissues between the MC and submesothelial blood vessels. Some submesothelial lymphatic vessels are attached to small holes in the MC and mesothelial cell layer (**left**), *, which are named lymphatic stomata (**right**). Scanning electron microscopic study was performed by Miura M, one of the authors of the article, and the techniques of SEM are described in reference [16].

**Figure 2 jcm-11-00458-f002:**
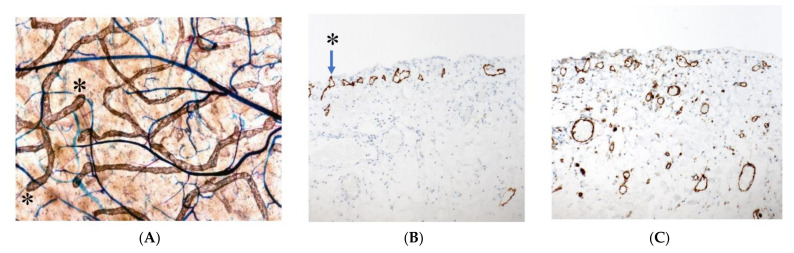
(**A**) Whole-mount specimens of Morrison’s pouch stained by alkaline-phosphatase (ALP) and 5′-nucleotidase double enzyme staining method show subperitoneal blood capillaries (blue) and lymphatic vessels (brown). (**B**) Lymphatic vessels and initial lymphatic vessels (*) stained with D2-40 monoclonal antibody. (**C**) Blood vessels stained with anti-CD31 monoclonal antibody. The specimens were obtained from macroscopically normal Morrison’s pouch from patients with gastric cancer. (**A**) ALP-5′-Nase double staining. * shows initial lymphatic vessel. (**B**) D2-40 staining shows submesothelial lymphatic vessels and initial lymphatic vessels. (**C**) CD31 staining shows submesothelial blood vessels.

**Figure 3 jcm-11-00458-f003:**
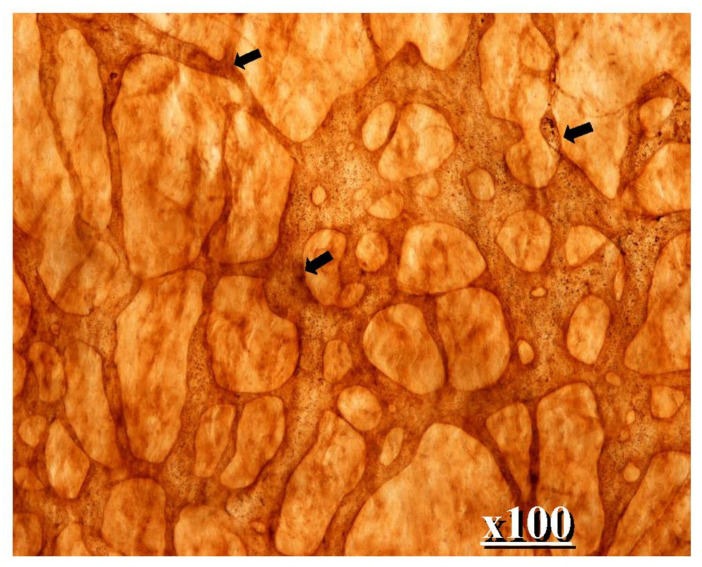
CH44 in submesothelial lymphatic vessels (arrow) of Morrison’s pouch.

**Figure 4 jcm-11-00458-f004:**
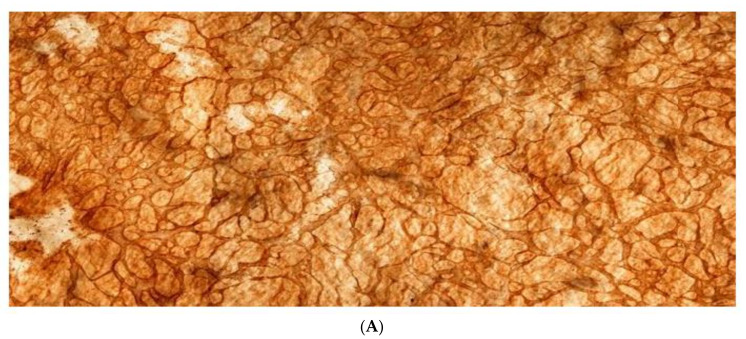
Lymphatic vessels in a falciform ligament. (**A**) (upper). Submesothelial lymphatic vessels in a falciform ligament (5′-Nase staining). (**B**) (lower left). Stomata on falciform ligament (arrow). (**C**) (lower right) Submesothelial lymphatic vessels in a falciform ligament (D2-40 immuno staining).

**Figure 5 jcm-11-00458-f005:**
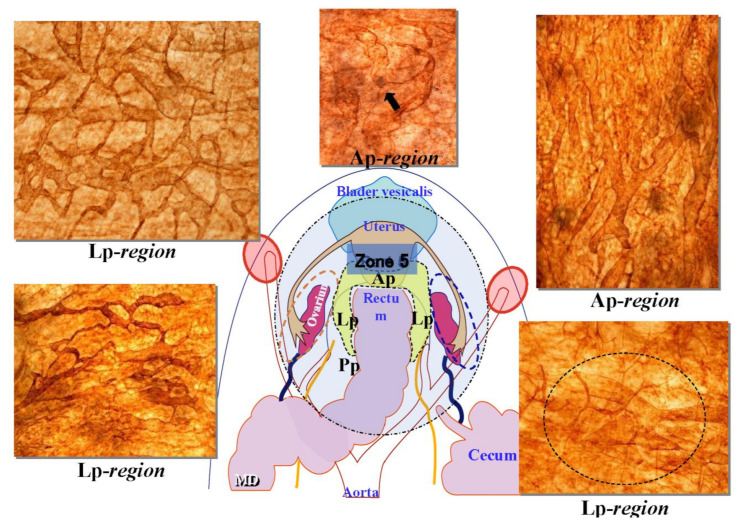
Submesothelial lymphatic vessels in human peritoneum (5′-Nase staining).

**Figure 6 jcm-11-00458-f006:**
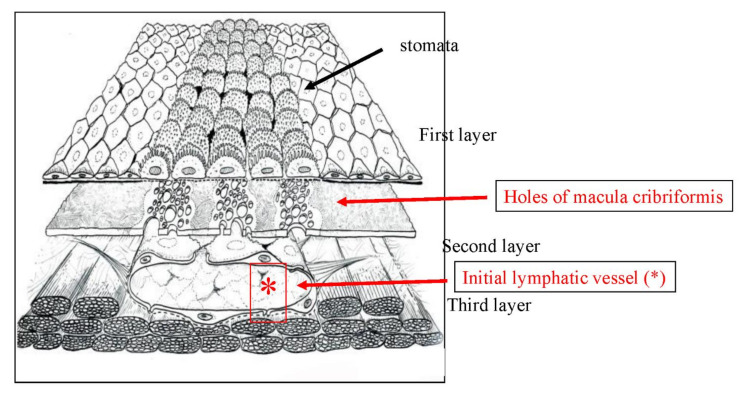
Schematic showing the triplet structure of the peritoneum in Morrison’s pouch, the para-colic gutter, and the pelvic peritoneum. The first layer is the mesothelial cells and basement membrane. The second layer is a sieve-like collagen plate named macula cribriformis (MC). The third layer is initial lymphatic vessels (*) attached to holes of MC.

**Figure 7 jcm-11-00458-f007:**
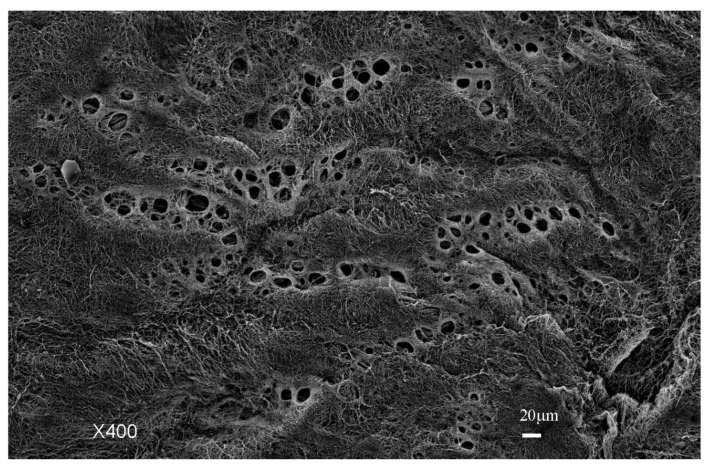
Collagen plate with small holes (MC). Below the submesothelail basement membrane, the MC contains multiple holes, and a blind loop of a submesohelial lymphatic vessels is attached to hole (scanning electron micrography taken after maceration by 6N KCL. The specimen is Morrison’s pouch from gastric cancer patient. Diameters of the holes of MC ranged from 30 μm to 150 μm.

**Figure 8 jcm-11-00458-f008:**
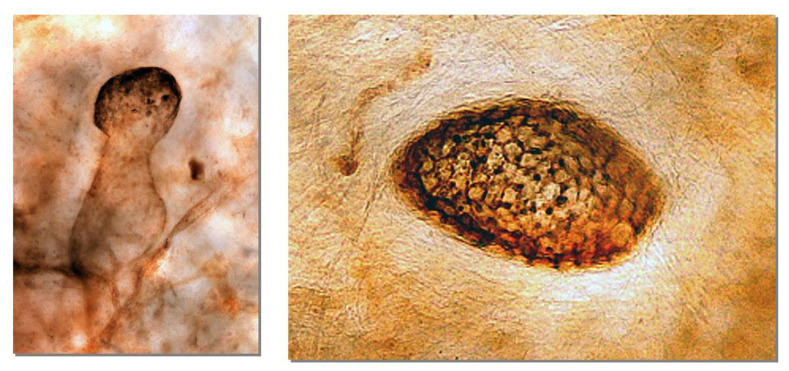
Blind loop of submesothelial lymphatic vessels (initial lymphatic vessel, stained with 5′-Nase enzyme) containing activated carbon (CH44) that was intraperitoneally injected (**left**). CH44 is attached between the lymphatic endothelial cells (**right**). (Pelvic peritoneum).

**Figure 9 jcm-11-00458-f009:**
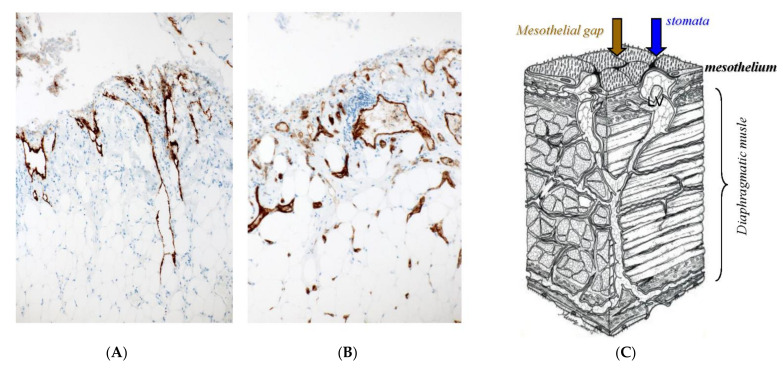
Lymph–vascular system of the diaphragm and the three-dimensional structure of the lymph–vascular network in the diaphragm. (**A**) D2-40 immunostaining. (**B**) CD31 immunostaining. (**C**) Schema of the diaphragmatic lymphatic system.

**Figure 10 jcm-11-00458-f010:**
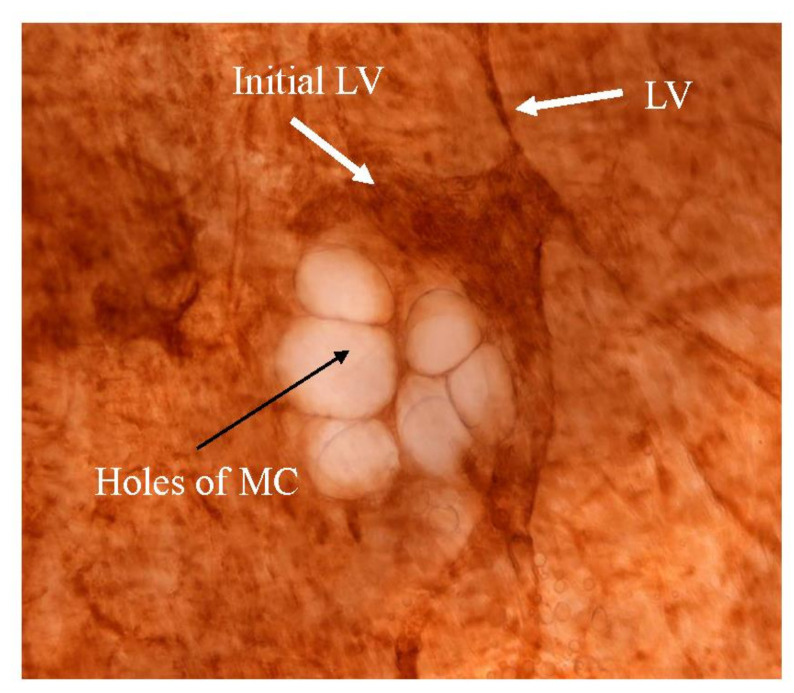
Holes in the diaphragmatic macula cribriformis (MC) connect with the initial lymphatic vessels (5′-Nase enzyme staining). Initial LV: initial lymphatic vessel. LV: lymphatic vessel connects with the initial lymphatic vessel.

**Figure 11 jcm-11-00458-f011:**
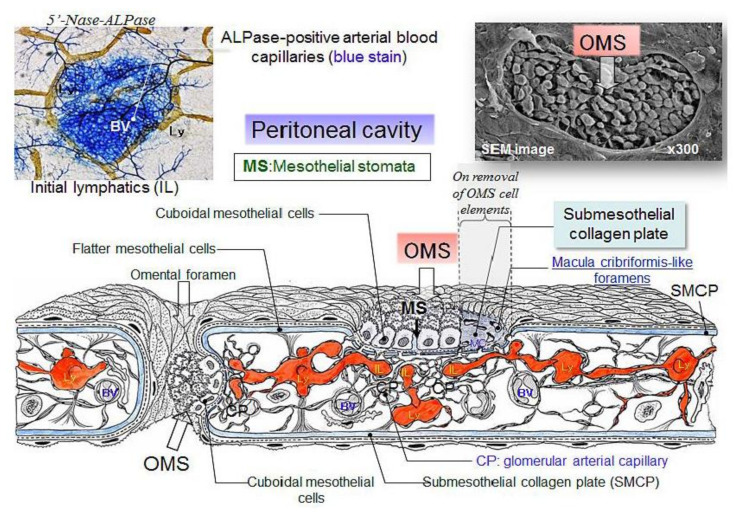
Schematic of the structures of omental milky spots (OMS). OMS are round or oval-shaped organs, covered with cuboidal mesothelial cells (right upper). Between the cuboidal mesothelial cells, stomata are connected to holes in the MC. Below the hole in the MC, there are glomerular-like blood capillaries and initial lymphatic vessels. (upper left). Ascites and inflammatory cells migrate from stomata to lymph–vascular system of the OMS through holes in the MC. Efferent lymphatic fluid absorbed from the OMS drain into the regional lymph nodes of the stomach.

**Figure 12 jcm-11-00458-f012:**
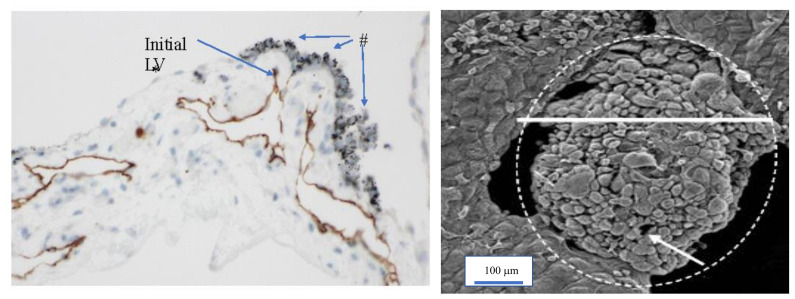
Histological features of human OMS stained with D2-40 antibody. The surface of the OMS is covered with cuboidal mesothelial cells. Macrophages (#) engulfing CH44 are located between the cuboidal mesothelial cells. Cuboidal mesothelial cells are surrounded by flat mesothelial cells (*) (**left**). The OMS activated by peritoneal carcinomatosis appear elevated due to the accumulation of macrophages on the surface (**right**).An arrow shows stoma connecting to holes in the MC and initial lymphatic vessels (**right**).

**Figure 13 jcm-11-00458-f013:**
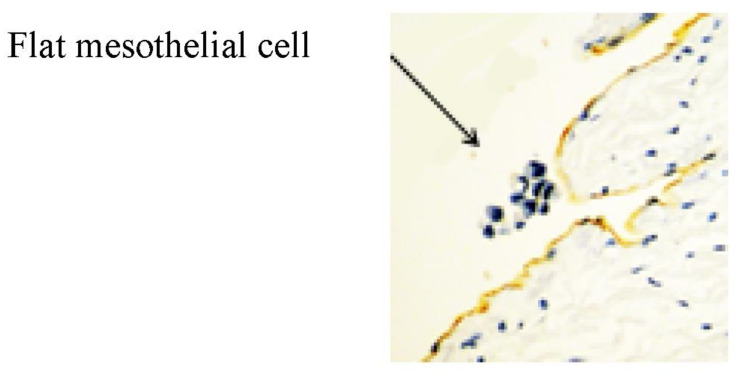
Cluster of cancer cells attached to mesothelial cells of the pelvic peritoneum that was obtained from gastric cancer patients by peritonectomy (D2-40 immune staining).

**Figure 14 jcm-11-00458-f014:**
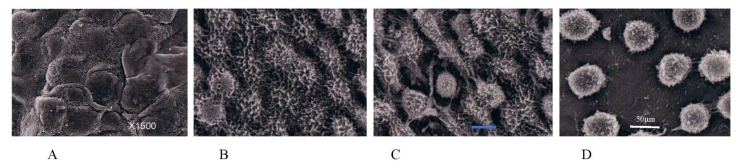
Morphological changes in mesothelial cells. (**A**) Normal mesothelial cells are flat and cuboidal in shape and attach to each other without inter-cellular gap. (**B**) Mesothelial cells activated by PFCCs show shrinkage and have well-developed microvilli. (**C**) The mesothelial cells separate from each other, exposing basement membrane as peritoneal metastasis progresses. (**D**) Finally, as mesothelial cells become round in shape, the submesothelial basement membrane becomes widely exposed.

**Figure 15 jcm-11-00458-f015:**
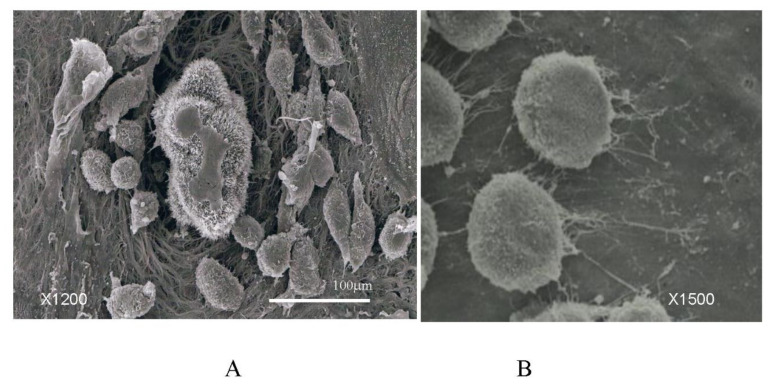
(**A**) Cluster of cancer cells attach to areas of the basement membrane exposed by shrinkage of mesothelial cells. (**B**) Micro villi extending from PFCCs attach to the basement membrane of the greater omentum. SEM finding of human gastric cancer cell line of MKN-45 that was co-cultured with human greater omentum for 2 h.

**Figure 16 jcm-11-00458-f016:**
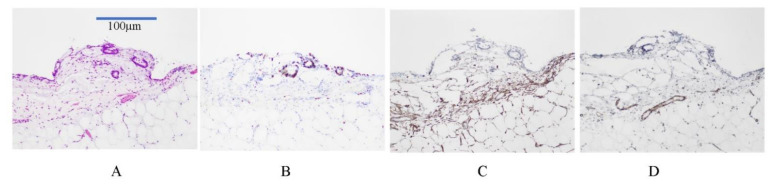
(**A**) Micrometastasis of differentiated adenocarcinoma growing on the small bowel mesentery (HE staining). Stromal induction is already found. The specimens are obtained from macroscopically normal paracolic gutters of patient with gastric cancer. (**B**) Ki67 staining shows positive immunoreactivity in cancer cells. (**C**) CD34 immunoreactivity is found at the front of micrometastasis invasion. (**D**) No angiogenesis is found in the stroma of micrometastasis (CD31 immunostaining).

**Figure 17 jcm-11-00458-f017:**
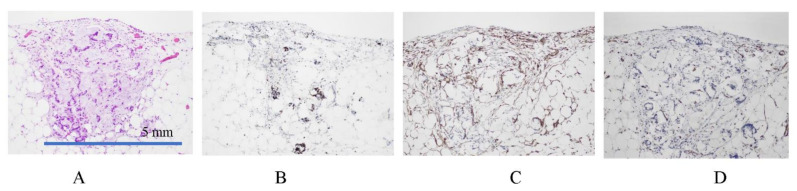
(**A**) An established metastasis of differentiated adenocarcinoma growing on the small bowel mesentery (HE staining). Stromal induction is already found. (**B**) Ki67 positive immunoreactivity in cancer cells. (**C**) CD34 immunoreactivity is found in the stroma of metastasis. (**D**) Newly formed blood vessels are found in the stroma (CD31 immunostaining).

**Figure 18 jcm-11-00458-f018:**
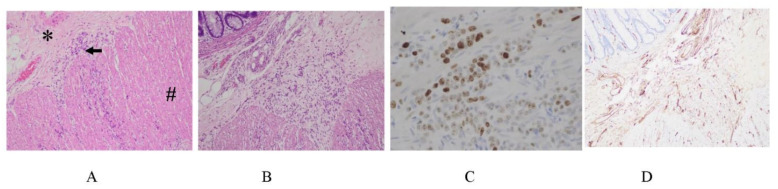
(**A**) Micrometastasis of poorly differentiated adenocarcinoma (arrow) growing on the subserosal layer of the rectum in Douglas pouch (*) and muscle layer of the rectum (#) (HE staining). Stromal induction is already found. (**B**) Poorly differentiated adenocarcinoma invading the submucosal layer. (**C**) Ki67-positive cancer cells in the submucosal layer. (**D**) Interstitial tissue of micrometastasis in the submucosal layer shows CD34 immunoreactivity.

**Figure 19 jcm-11-00458-f019:**
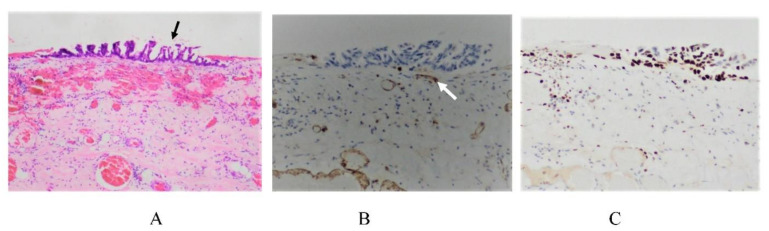
(**A**) Superficial growing metastasis. Micrometastasis of differentiated adenocarcinoma (arrow) growing on the paracolic gutter (HE staining). (**B**) CD31-positive submesothelial blood vessels (arrow) are detected just below the micrometastasis. (**C**) Ki67-positive cancer cells in the micrometastasis.

**Figure 20 jcm-11-00458-f020:**
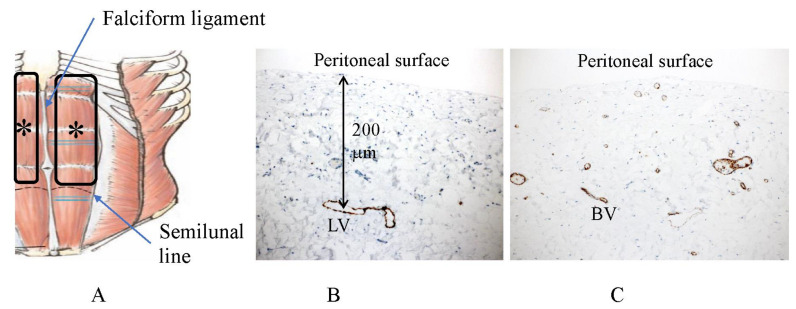
Subperitoneal blood vessels (**C**) in the peritoneum of the anterior abdominal wall extending from the hypochondrium to the semilunal line except the falciform ligament (*) (**A**) (CD31 immunostaining). (**B**) Subperitoneal lymphatic vessels in the peritoneum of anterior abdominal wall (D2-40 immune staining) located 200 μm below the peritoneal surface. (**C**) Density of submesothelial blood vessels in the anterior abdominal wall is scarce.

**Figure 21 jcm-11-00458-f021:**
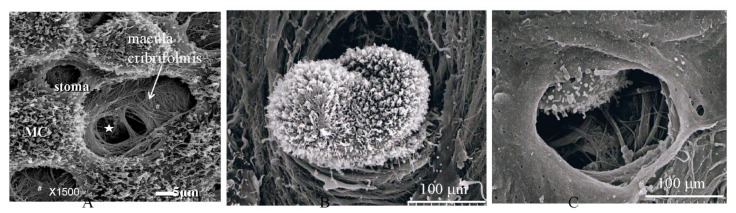
(**A**) Holes (*; stoma) in the macula cribriformis revealed between shrunken mesothelial cells (MC). Initial lymphatic vessels (Figure 6) exist just below the stomata. (**B**) PFCCs are attached to the stoma. (**C**) PFCC invade into the initial lymphatic vessel. Diameter of the hole of MC is 125 μm. MC: macula cribriformis.

**Figure 22 jcm-11-00458-f022:**
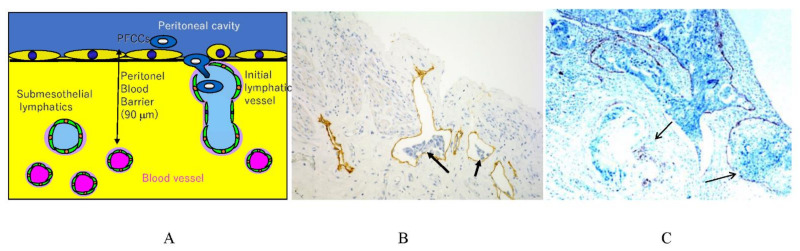
(**A**) Schema of trans-lymphatic metastasis. PFCCs invade the initial lymphatic vessels through mesothelial stomata and hole in the macula cribriformis. (**B**) Cancer cells (arrows) proliferating in the lymphatic lacuna connecting with initial lymphatic vessels (D2-40 immunostaining). (**C**) Lymphatic vessels are destroyed by the proliferation of cancer cells (arrows, D2-40 immunostaining), resulting in the malfunction of lymphatic vessels.

**Figure 23 jcm-11-00458-f023:**
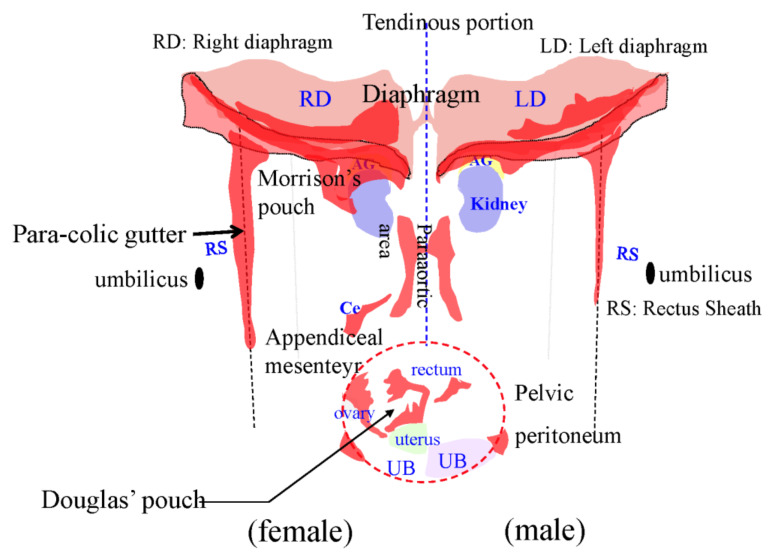
Schematic shows the initial lymphatic vessel-rich peritoneal sectors (pink-coloured sectors).

**Figure 24 jcm-11-00458-f024:**
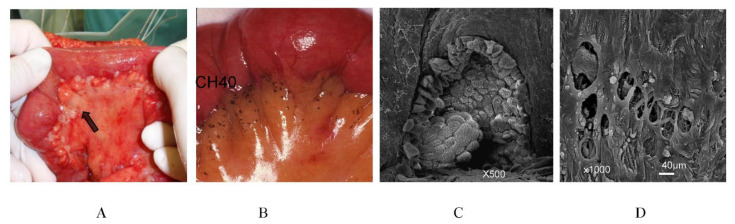
(**A**) Metastasis from gastric cancer on the small bowel mesentery. Metastatic nodules are found at the site of the mesenteric attachment to the small bowel. (**B**) CH44 intraperitoneally injected is present in similar sites of metastasis to those in (**A**). (**C**) Scanning electron micrography shows the omental milky spots (oval shaped-structure) covered with cuboidal mesothelial cells. Between the cuboidal mesothelial cells, stomata-like holes are observed. (**D**) After maceration with KOH, the macula cribriformis with holes is detected.

**Figure 25 jcm-11-00458-f025:**
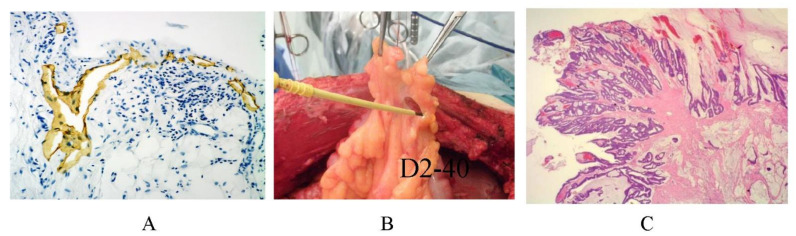
Initial lymphatic vessels are found in the epiploic appendage of the colon (**A**). (**B**) Resection of the epiploic appendage. (**C**) Histologic features of metastasis on the epiploic appendage.

**Figure 26 jcm-11-00458-f026:**
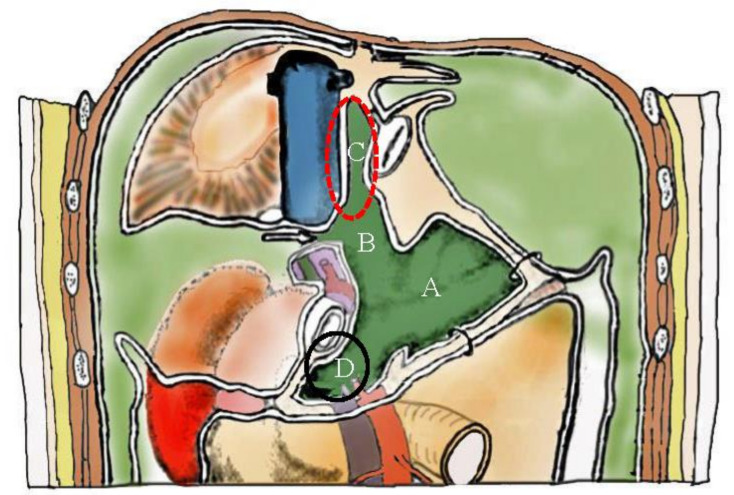
The omental bursa surrounded by the anterior leaf of the transverse mesocolon, pancreas capsule, posterior wall of the stomach, inferior vena cava, and the right crural muscle of diaphragm. Omental bursa opens into the peritoneal cavity through the foramen of Winslow, and consists of the greater omental sac (**A**), lesser omental sac (**B**), superior recess (**C**), and anterior vestibule (**D**).

**Figure 27 jcm-11-00458-f027:**
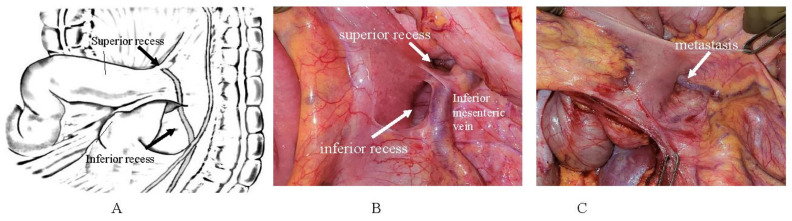
(**A**,**B**) Duodenojejunal junction has two pockets, i.e., the superior recess and inferior recess. (**A**,**B**) Duodenojejunal junction has two pockets, i.e., the superior recess, and inferior recess. (**C**) Metastasis on the inferior recess.

**Figure 28 jcm-11-00458-f028:**
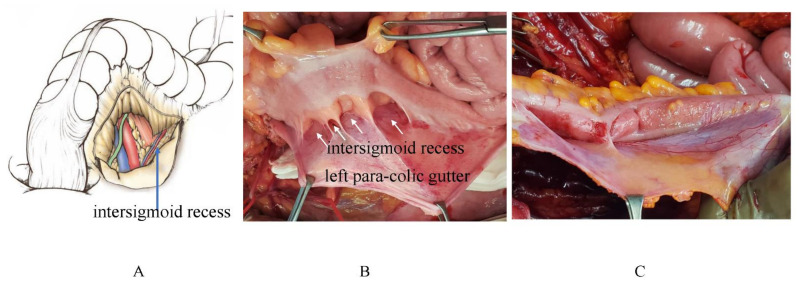
(**A**,**B**) Left side of the sigmoid mesocolon has an intersigmoid recess. (**B**) This case has four intersigmoid recesses. (**C**) Metastases on the intersigmoud recess and left para-colic gutter are removed by peritonectomy.

**Figure 29 jcm-11-00458-f029:**
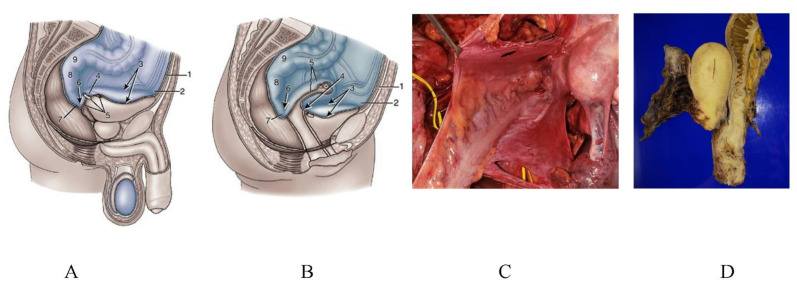
(**A**) Anatomical structure of the pelvic organs and peritoneum in males. (**B**) Anatomic structure of the pelvic organs and peritoneum in females. (**C**) Resection of a metastasis on a Douglas pouch by peritonectomy. (**D**) Cut surface of the resected specimen by posterior pelvic exenteration.

**Table 1 jcm-11-00458-t001:** Drug molecular weights, pAUC/sAUC, penetration distance into the subperitoneal tissue after intraperitoneal administration, maximum tolerate doses (MTD) and thermal enhancement with each drug. NA: not analyzed, pAUC: area under the curve in peritoneal cavity, sAUC: area under the curve in serum [10,12].

Drugs	MW	pAUC/sAUC	Penetratoin Distance	MTD	Thermal Enhancement
Doxorubicin	380	230	4–6 cell layer	15 mg/m^2^	Yes
Melphalan	305	93	NA	70 mg/m^2^	Marked
Mitomycin C	334	32.5	NA	35 mg/m^2^	Yes
Cisplatin	300	7.8	1–2 mm	300 mg/m^2^	Yes
Gemcitabine	299	500	NA	1000 mg/m^2^	at 48 h
Miroxantron	517	115–255	5–6 cell layer	28 mg/m^2^	Yes
Oxaliplatin	387	16	1–2 mm	460 mg/m^2^	Yes
Etoposide	568	63	NA	200 mg/m^2^	Yes
Irrinotecan	677	NA	NA	NA	No
Paclitaxel	853	10,000	80 cell layer	120–180 mg/m^2^	No
Docetaxel	861	552	1.4 mm	156 mg/m^2^	Yes
5-FU	130	250	0.2 mm	650 mg/m^2^ × 5 days	Yes
carboplatin	371	10	0.5 mm	300 mg/m^2^	Yes

## Data Availability

The photographs and figures shown in this study was totally from the study by our selves. The study did not report anywhere.

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
