# Peer review of "The Development of Peritoneal Metastasis from Gastric Cancer and Rationale of Treatment According to the Mechanism"

_jcm, 2022, doi:10.3390/jcm11020458_

Round 1
Reviewer 1 Report
The reviewer appreciates the effort the authors put in this paper, which shows the variety of methods established in the lab. However, the maunscript can be improved.
It is not clear why the paper is submitted under the category of commentary and it should be considered to submit it as a review. The title is misleading and not well chosen for the presentation of anatomy of the peritoneum and the development of peritoneal metastasis in gastric cancer.
The introduction needs to be re-written as it contains disrupting thoughts and doubling of facts.
The visual presentation is well done, apart from figure 13, and maybe the nz,ber of pictures could be reduced to the relevant ones ( 30 figures are a lot for one paper).
Further, the re-structuring of the paper and focus on one theme would improve the attractivity of the paper.
Author Response
Answers to Reviewer 1
Thank you for your review
1; I wrote the manuscript as a review, I changed the title as Development of peritoneal metastasis from gastric cancer and rationale of treatment according to the mechanism,
according to your suggestion.
2;Introduction is re-wretten, according to your comment.
As followings;
Introduction
Gastric cancer (GC) is the second leading cause of deaths from cancer. Peritoneal metastasis (PM) is the most common forms of metastasis in GC and PM is found about 14% of primary GC cases (1,2). However, patients have a median survival time of 3-6 months (1,2). Until the early 1990s, GC with PM was considered an incurable disease, because it could not be cured by surgery or systemic chemotherapy alone (3,4). Complete removal of PM, disseminated to the peritoneal surface cannot be achieved by standard surgical techniques. Even after complete resection of a small number of PM with gastrectomy plus lymph adenectomy, residual micrometastasis on the peritoneal surface always proliferate and almost all patients will die. Systemic chemotherapy using modern drugs has limited effects on PM (3,4), because only small amounts of systemically administered drugs can enter the peritoneal cavity, and even effective regimens are inevitably interrupted due to the development of side effects or regrowth of multidrug-resistant cancer cells.
In the late 1990s, a combination of cytoreductive surgery (CRS) and perioperative chemotherapy for the treatment of PM was proposed as a comprehensive treatment by the Peritoneal Surface Oncology Group International (PSOGI) (5). In CRS, all the macroscopically detectable tumors including the primary tumor, lymph node metastasis and peritoneal nodules are removed by D2-gastrectomy and peritonectomy (5,6). However, even macroscopic complete cytoreduction leaves invisible micrometastasis in most cases (7). To eradicate the micrometastases before and after CRS, neoadjuvant IP chemotherapy and intraoperative hyperthermic intraperitoneal chemoperfusion (HIPEC) were developed (8-11).
The comprehensive treatment improved the long-term survival of GC patients with PM and 10% of patients survived loner than 10 years (11), Accordingly, the treatment is now considered a curative approach (6,9,11). For the development of more effective treatments to improve survival, it is important to clarify the mechanisms of the formation of PM.
The present chapter presents the mechanisms of PM from GC, and the rationale for eliminating micrometastasis by chemotherapy.
3 I reduced several figures; Figure 15-c and Figure 30 were omitted. However it is very difficult reduce other Figures to completely explain the mechansims of the formation of PM.
.
4+The MS consists of two parts, i.e., basic research and the pits for clinical application to succeed the methods to eradicate micrometastasis on the peritoneal surface. Only a few studies have been reported about the mechnisms of PM in human gastric cancer. Accordingly, the main contents of the MS in the basic researches came from our original studies except molecular biological researches. Knowledges of the basic mechanisms of the formation of PM is very important to succeed neoadjuvant chemotherapy (NAC).
If the micrometastasis burden is less than the threshold levels after NAC that could be completely eradicated by intraoperative HIPEC, patients could be cured.
Accordingly, these two parts cannot be separated and the part of clinical application cannot be omitted. So, I want to publish our MS in the present form.
Reviewer 2 Report
Whilst the paper is well written and contains excellent images and diagrams, it suffers from some methodological flaws. There is no information supplied as to whether some of the 'normal' peritoneal specimens shown in the manuscript have been removed from normal human patients or what operations they were having, or whether these are post mortem specimens. There is no ethical statement about whether consent has been received for specimen removal, or whether the specimens were obtained from a tissue biobank, or approval obtained for the study from the local IRB. This requires correction. There is minimal discussion of the results of RCTs in the efficacy of HIPEC/CRS vs CRS in gastric cancer PM. There is no discussion on the influence of Peritoneal cancer index (PCI) on the prognosis of the patient or how this influences treatment modalities. There is no discussion on future potential treatments for PM eg immunotherapy, viral delivery of anti-cancer agents, nanomolecules, next generation sequencing in molecular genetics, PIPAC etc. There is no conclusion section. As such, this limits the usefulness of the paper for the reader.
Line 255 characteristically spelling
Line 71 Need to insert a methods section
Line 72 What is the role of ascites in peritoneal metastasis?
Line 87 Are water and electrolytes adsorbed? Please correct.
Line 91 What are the size of the holes in um, as compared to the diameter of cancer cells or host macrophages?
Line 106 A short discussion on the movement of ascites fluid in the peritoneal cavity and the reasons why peritoneal metastases often appear first on the surface of the diaphragm ie effects of inspiration/expiration on ascites movement; and also peritoneal stomata on the diaphragm.
Line 158 Figure 8 should possibly have labels on the diagram to improve its interpretation
Line 200 What is the role of loss of E-cadherin in EMT in gastric cancer?
Line 211 What size of HA is important? HA fragments enhance cancer cell attachment, but high molecular weight HA (HMWHA) does not. What is the role of IL-1/IL-1R, IL-6, integrins, CD44 and RHAMM in attachment to peritoneal mesothelial cells?
Line 225 What about reactive oxygen species/lactate/EGF from cancer cells in changing behaviour of mesothelial cells? What is the contribution of the host stromal cells to the growth of peritoneal metastases ie recruitment of host stroma? How does this relate to tissue hypoxia, VEGF and angiogenesis?
Line 241 What other mechanisms cause exposure of the peritoneal basement membrane eg surgical trauma, cold dry CO2 pneumoperitoneum?
Line 244 What is the role of the mesothelial glycocalyx in protecting the mesothelial cells from cancer cells/damage?
Line 281 What are the motility factors (eg. EMT transcription factors (SNAIL,SLUG,TWIST), N cadherin, vimentin), adhesion molecules (eg integrins), and matrix digesting proteins (eg. MMP7, MMP9)?
Line 283 What is the contribution of TGF-B and VEGF sequestered in the extracellular matrix, which is released by MMP/lactate?
Line 375 How do cancer cells migrate through the peritoneal cavity-via transport by flow kinetics (eg. ascites) or their own movement (amoeboid/rolling)?
Line 404 What about peritoneal blood flow (100ml/min) limiting systemic chemotherapy delivery to the peritoneum- is that important?
Line 414 spelling of docetaxel
Line 422 What chemotherapy drugs penetrate a metastasis with high interstitial pressures and by how far? Is this important in large scirrhous PM nodules? Does this influence the effectiveness of IP chemotherapy?
Line 428 What are the limitations of standard peritoneal chemotherapy versus pressurized intraperitoneal chemotherapy (PIPAC)?
Line 431 I suggest references are provided for this table. By how much does hyperthermia enhance the effect of IP docetaxel (as a tubulin binding agent)? How does this compare to DNA damaging agents such as doxorubicin and cisplat which are synergistic with hyperthermia in inhibiting DNA repair?
Line 434 Need to define NIPS: neoadjuvant intraperitoneal-systemic chemotherapy protocol (NIPS)
Line 453 There are no conclusions. Need to discuss how the macro- and microanatomy of peritoneal metastases influences the delivery and effectiveness of IP chemotherapy and the reasons why IP chemotherapy can fail to control PM ie. patterns of chemotherapy resistance in gastric cancer PM.
Line 526 Need to correct year reference 36 was published
Line 634 The format of reference 90 requires correction
Author Response
Answers to Reviewer 2
Thank you for your review
1; suspension means the medium dispersed cells.
2; I introduced paragraphs about methods, as followings, Macroscopically normal peritoneal parts were obtained from the resected specimens of GC patients for performing peritonectomy, and studied the structures of lymphatic vascular system by immunohistochemistry and double enzyme staining method using alkaline-phosphatase (ALP) and 5’-nucleotidase (Nase) reactions.
The study was approved by an ethical committee of Kishiwada Tokusyukai Hospital as a study number 19-35, entitled “a clinical study of the efficacies of a comprehensive treatment of peritoneal metastasis”.
3; The function of normal ascites (peritoneal fluid) are described in line 74-76.
4; line 87; I changed the sentences as followings; Peritoneal fluid containing electrolytes is absorbed ---------
5: The diameters of the holes range from 20 to 150 mm. The graduations are shown in Fig,3B, Figure 21,B,C, and Figure 24, D in the revised version of Figures in PDF file.
6; Line 106; The reason why the PM often appear on the diaphragmatic surface is that peritoneal free cancer cells are trapped on the diaphragmatic lymphatic stomata by the negative pressure of respiratory inspiration, and migrate into the initial lymphatic vessels by their motility activities. The same sentences are described in line 110-114
7 Line 158; I add the labels to indicate initial lymphatic vessel (*) and holes of macula cribriformis.
in Figure 8.
8; Line 200; About the role of E-cadherin. I add the sensences ; E-cadherin expressed on the adherence junction has a major role in the cell-cell adhesion. The loss of cadherin induces loosening the adhesion, resulting in the dispersion of cells.
9 I donot know the precise mechanisms of PFCC attachment on mesothelial cells via LMWHA. Maharjan AS et al reported that LMWHA stimulates macrophage to secrete inflammatory cytokines such as IL-8. Additionary, Amorim S et al, reported that LMWHA activate the cytoskeleton rearrangement, and promote cellular mortility and signaling pathway on gastric cancer cells (Adv Biposyst 2020)/
IL-1/IL-1R, IL-6 induce mesothelial contraction, resulting in the exposure of submesothelial basement membrane. Integrins expressed on PFCCs have a big role in the heterotopic adhesion of cancer cells and the elements of basement membrane. VLA-2 and VLA-3 (Ref 44, 45) overexpressing GC cell line is known in a highly metastatic GC cell line on the peritoneal surface. The evidences are cited from Ref. 44, 45, 46.
CD44 expressed on PFCCs is the ligand of hyaluronate, and E-selectin on the mesotherial cells.
The receptor for hyaluronic acid-mediated motility (RHAMM) is known to upregulated in various cancers. In pancreatic cancer, it may serve as a prognostic facter and hemtogenous metastasis. Amorim S et al reported that LMWHA trigger the invasive activity of gastric cancer cells through the HA receptor RHAMM.
10, I do not know the data reactive oxygen species/ lactate/EGF from cancer cells in changing behavior of mesothelial cells. However, in our experiments, EGF induces polymerization of myosin, resulting in the morphological change of mesothelial cells. I have no idea about the contribution of the host stromal cells to the growth of PM, and tissue hypoxia ,VEGF and angiogenesis. I added the sentences as followings ; These cytokines are produced from not only cancer cells but host inflammatory cells and fibroblasts (39-42).
11; Line 241; Surgical trauma, cold dry and CO2 pneumo peritoneum may contribute to the basement membrane exposure. However, I have no data.
- Line 244 have no data of mesothelial glycoclyx in protecting the mesothelial cells from cancer cells/damage.
13 Line 281, concerted expressions of motility factors, adhesion molecules, like VLA families, and matrix digesting enzymes may associate with the invasion of poorly differentiated adenocarcinoma.
- Line 283, TGF-B and VEGF produced from fibroblasts and cancer cells must associated with angiogenesis .
- PFCCs migrate into the pocket-like structures via peritoneal fluid. During rolling in the pockets, adhesion molecules like CD44, P-cadherin, and some of integrin molecules (a4b1, a4b7, a1b2, a4b7 expressed from PFCCs and their receptor, hyaluronate, S-selectin, P-cadherin, ICAM-1, VCAM may associate with the lodging of peritoneal free cancer cells on peritoneal surface of the pockets.
- Of course, systemic chemotherapy brings drugs into the subperitoneal tissue, but the amounts delivered by systemic chemotherapy are limited, as compared to those after IP chemotherapy.
17.docetaxel is correct, and I changed.
- There have been no report about the penetration distances of anticancer drugs in human beings. In the scirrhous cancer supposed to have a high interstitial pressure, repeat intraperitoneal administration of drugs may overcome the difficult point.
- Drug penetration distance by PIPAC may be longer than the simple IP chemo. However, PIPAC have limiting factors i.e., 1) repeat administration is difficult, 2) amounts of drugs are small, 3)
Drugs introduced by PIPAC do not reach to cancer cells proliferating in the deep subperitoneal tissue, and 4) administered drugs are excreted to normal tissue due to the high interstitial pressure in the peritoneal metastasis.
- Referrences are from Ref 10, 12. Doxorubicin, MMC and CDDP are known to have synergistic effect with heat treatment. But deBree reported that “The concentration-dependent cytotoxic effect of docetaxel supports their intraperitoneal use. Due to the lack of or only minimal thermal enhancement, normothermic may be as effective as hyperthermic intraoperative intraperitoneal chemotherapy with taxanes” (de Bree E, Katsougkri D, Polioudaki H, Tsangaridou E, Michelakis D, Zoras O, Theodoropoulos P. Anticancer Res. 2020 Dec;40(12):6769-6780. doi: 10.21873/anticanres.14700. Epub 2020 Dec 7.).
In contrast, Mohamed F described that Docetaxel shows a moderate increase in anti-tumor activity with hyperthermia. At 41.5 degrees C the thermal enhancement of docetaxel is time dependent if hyperthermia is applied immediately following drug administration.
In our experiment, docetaxel reached to 1.4 mm from the peritoneal surface during 40 min. HIPEC.
Docetaxel binds with carrier vehicle and gradually release from the carrier. Hyperthermia may increase the penetration amounts of docetaxel in the peritoneal metastasis against high interstitial pressure.
- In NIPS, intraperitoneal administrations of 40mg of docetaxel and ciclatin with 500 ml of normal saline are performed on day 1 and 14, and oral intake of 60mg/m2 of S1 start from day 1 to day 14. After 1 week rest (from day 15 to day 21), NIPS repeats at least 3 cycles. One month after the last cycle of NIPS, patients opt for performing cytoreductive surgery received gastrectomy plus D2 lymph adenectomy and peritonectomy.
- I added conclusion paragraphs.
- years is 2021
- I corrected the format of Ref 90
Reviewer 3 Report
I congratulate the authors on an excellent and comprehensive review of Gastric peritoneal carcinomatosis. Few suggestions for authors' consideration
- Considering the importance of NIPS, it will be helpful to add additional information about the chemotherapy drugs typically used in this approach.
- Please add a conclusion paragraph.
Author Response
Reviewer 3
Thank you for your review
I added the information about NIPS.
Conclusion paragraph was added.
Round 2
Reviewer 2 Report
The title of the paper is ‘Mechanisms of peritoneal metastasis from gastric cancer and rationale of treatment’. The discussion of the mechanisms of cancer cells adherence to the mesothelium and invasion is still inadequate-the authors have only briefly mentioned interleukin 1, RHAMM, HA , integrins, VEGF, TGF-B which enable peritoneal metastasis and ascites formation. The authors have still not included a formal Methods section. It is unclear as to whether the peritoneal specimens were from patients having primary gastric cancer resections or whether it was during gastric cancer PM CRS/HIPEC after NIPS. The authors need to describe the techniques used to prepare the images, including scanning electron microscopy. In the SEM images, the magnification needs to be included (eg 2000x). I suggest the NIPS protocol be included in the methods section or appendix. There is very little discussion on the studies that show CRS/HIPEC is superior to CRS in peritoneal metastases, or the influence of the peritoneal cancer index (PCI) on the patient’s prognosis. There is little discussion on the literature evidence for the effectiveness of intraperitoneal chemotherapy in gastric PM in allowing subsequent CRS. There is no discussion on future potential treatments for PM eg immunotherapy, viral delivery of anti-cancer agents, nanomolecules, next generation sequencing in molecular genetics, PIPAC etc. As such, this limits the usefulness of the paper for the reader.
Corrections by line:
Line 91: The peritoneal glycocalyx is made up of surfactant and HMWHA and is produced by the mesothelial cells. This is what primarily provides the lubrication for viscera to slide over the visceral and parietal peritoneal surfaces without friction. Please correct.
Line 104 adsorbed and absorbed are 2 different mechanisms
Line 112 adsorbed
Line 140 Is it true that the omental milky spots are similar to that (or those) of the parietal peritoneum (PP), or is it the PP on the diaphragm/falciform ligament rather than the anterior PP?
Line 162 ranged from
Line 300 Is it 100mm or 100 microns (um)?
Line 326 adsorbed
Line 358 Fig 3 is pixellated-please provide a higher resolution image
Line 448 Please define the blood peritoneal distance as BPD. Is it really 50mm? What is the real distance from the peritoneal capillary to the mesothelial cell layer?
Line 458 The pharmacology of IP chemotherapy agents need to be scientifically discussed. For example what is the half-life of a HMW molecule such as docetaxel in the peritoneal cavity compared to cisplat? This then provides the basis for understanding the dosage schedule used in NIPS.
Author Response
Answers to Reviewer 2:
Thank you for advises.
The peritoneal specimens were from macroscopically normal peritoneum from patients who received cytoreductive surgery for gastric cancer or other diseases. Specimens are not from patients treated with NIPS.
2) Scanning microscopic study was performed by one of the authors of the MS, Professor Masahiro Miura, who is a specialist of anatomy. The techniques were described in the following article, Ushiro H, Miura M, Iobe H., et al. Lymphatic stomata in the adult human pulmonary ligament. Lymphat Res Bio, 2015, 13 (2), 137-145, doi 10.1089/lrb 2014 0009
I added the article as a reference No 91, which is sited in Figure legend of Figure 1.
3) I introduced scales in each SEM photograph.
4)NIPS protocol was described from line 483 to 486.
5) I added a new chapter 5
6)Effects of IP Chemo on PM from GC.
Effects of NIPS on PM from GC has been reported from Japanese surgical oncologists. Ishigami et al. conducted a randomized phase III trial to confirm the effects of NIPS. Patients were randomly assigned to two groups, i.e., 1) intraperitoneal and intravenous paclitaxel plus S-1 (IP; intraperitoneal paclitaxel 20 mg/m2 and intravenous paclitaxel 50 mg/m2 on days 1 and 8 plus S-1 80 mg/m2 per day on days 1 to 14 for a 3-week cycle) and 2) S-1 plus cisplatin (SP; S-1 80 mg/m2 per day on days 1 to 21 plus cisplatin 60 mg/m2 on day 8 for a 5-week cycle.
This trial failed to show statistical superiority on survivals of NIPS group as compared with control group. However, the response rate analyses suggested possible clinical benefits of NIPS for PM from GC.
Yonemura Y et al. studied the effects of NIPS by laparoscopy, and the peritoneal cancer index (PCI) was significantly reduced after 3 cycles of NIPS. These results indicate that IP chemotherapy is effective on PCI reduction. Additionally, Yonemura et al. reported that post NIPS PCI =<6 was the strongest independent prognostic factor.
1. Yonemura Y, Canbay E, Fujita T, Sako S, Wakama S, Ishibashi H, Hirano M, Mizumoto A, Takeshita K, Takao N, Ichinose M, Noguchi K, Liu Y, Li Y, Taniguchi K. Effects of neoadjuvant laparoscopic hyperthermic intraperitoneal chemoperfusion and intraperitoneal/systemic chemotherapy on peritoneal metastasis from gastric cancer. Journal of Peritoneum (and other serosal surfaces) [eISSN 2531-4270], Pavia, Italy.JoPer [Internet]. 31Jul.2017 [cited 5Jan.2019];2(2). Available from: http://www.jperitoneum.org/index.php/joper/article/view/60
I added a new chapter 7
7.. Effects of HIPEC on PM from GC
Yonemura et al studied the effect of laparoscopic HIPEC on PM from GC. Laparoscopic HIPEC (LHIPEC), receiving cisplatin 50mg and docetaxel 40mg in 4000 ml of normal saline at 43 ± 0.5°C for 60 min were performed in 55 GC patients with PM, and laparoscopy was again performed one month later. PCI was significantly reduced and the positive cytology became negative in 62% of patients with positive cytology at the time of LHIPEC.
Brandl A collected and analyzed a worldwide cohort of patients treated with cytoreductive surgery and HIPEC with long-term survival in order to explore relevant patient characteristics (93). From an analysis of 448 patients, a total of 28 patients with mean PCI of 3.3 were survived longer than 5 years after CRS plus HIPEC. The overall median survival was 11.0 years (min 5.0; max 27.9). The predictors completeness of cytoreduction (CC-0) and PCI<6 were present in 22/28 patients. They concluded that completeness of cytoreduction and low PCI seemed to be crucial, and that long-term survival and even cure are possible in patients with PM of GC treated with CRS and HIPEC.
Yang XJ performed a randomized phase III study to evaluate the efficacy and safety of CRS plus HIPEC for PM from GC (94). Sixty-eight gastric PC patients were randomized into CRS alone (n = 34) or CRS + HIPEC (n = 34) receiving cisplatin 120 mg and mitomycin C 30 mg each in 6000 ml of normal saline at 43 ± 0.5°C for 60-90 min. The median survival was 6.5 months in CRS and 11.0 months in the CRS + HIPEC groups (P = 0.046). Multivariate analysis found CRS + HIPEC, synchronous PC, CC 0-1, systemic chemotherapy ≥ 6 cycles, and no serious adverse events were independent predictors for better survival. They concluded that CRS + HIPEC with mitomycin C 30 mg and cisplatin 120 mg may improve survival with acceptable morbidity (94).
These results indicates that HIPEC is directly effective on PM from GC, and HIPEC may improve GC-patients’ survival after complete resetion of PM.
I added Future perspectives
Future perspectives
IP immunotherapy offers a novel approach for the control of regional disease of the peritoneal cavity by breaking immune tolerance. These strategies include heightening T-cells, vaccine induction of anti-cancer memory against tumor-associated antigens (95).
Catumaxomab, a nonhumanized chimeric antibody, is characterized by its unique ability to bind to three different types of cells: tumor cells expressing thr epithelial cell adhesion ,olecule (EpCAM), T lymphocytes (CD3) and also accessory cells (Fcy receptor). Because up to 90 % of gastric cancer express EpCAM, intraperitoneal infusion of catumaxomab after complete resection of all macroscopic disease could therefore efficientry treat macroscopic residual disease (96). Knödler M performed a prospective, randomised, phase II study investigated the efficacy of catumaxomab followed by chemotherapy (arm A, 5-fluorouracil, leucovorin, oxaliplatin, docetaxel, FLOT) or FLOT alone (arm B) in patients with GC and PM. However, no survival benefit was found in treatment group (96).
Nivolumab, anti-programmed cell death-1 (PD-1) antibody has been developed and survival benefit was obtained in GC-patients with PM (97). However, no randomized phase III study was reported.
Nab-paclitaxe (nanoparticle alubumin-bound paclitaxel) has effective transferability to tumor tissue and strong antiyumor effects for peritoneal metastasis (98). Ishikawa M reported that nanoparticle alubumin-boun (nab)-PTX treatment has beneficial effects on survival of GC-patients with PM as compared RAM plus solvent-based (sb)-paclitaxel (98).
Pressurised intraperitoneal aerosol chemotherapy (PIPAC) was introduced as a new treatment for patients with peritoneal metastases in 2011. Adverse events greater than grade 2 occurred after 12-15% of procedures. An objective clinical response of 50-91% for gastric cancer (median survival of 8-15 months). Alyami M et al. reported that PIPAC was safe and objective response and quality of life were encouraging. Therefore, PIPAC can be considered as a treatment option for refractory, isolated peritoneal metastasis (99).
Patients with PM may be improved by gene therapy (100). Many ideas has been reported, but clinical supplication has not been performed.
We are awaiting the developments of new effective nanomolecules, cancer-specific antibody or anti-cancer drugs.
In the surgical field, further studies are needed in order to improve existing selection criteria for cytoreductive surgery.
Line 91 I corrected the description of line 91 to the followings;
Ascites and the glycocalyx like hyaluronic acid produced from mesothelial cells plays a role as a lubricant for viscera to slide over the visceral and parietal peritoneal surface without friction.
Line 104; I changed adsorb to absorbed.
Line 112; I changed “CH44 is adsorbed” to “CH44 is absorbed”.
Line 140; You are wright. I changed the sentences to “The fundamental structure of OMS is similar to that of the parietal diaphragm/falciform ligament and te peritoneum of pelvis, para-colic gutter and Morrison’s pouch”.
Line 162; I changed to “ranged from”
Line 300. 100 mm is correct.
Line 326 changed to “PFCCs are adsorbed to the stomata between the cuboidal mesothelial cells covering the OMS, and invade into the omental lymphatic vessels through holes in the MC (Figure 11,12).”
Fogure 13 is changed, Please do not magnify the Figure.
Line 448: 90 mm was changed to 90mm
Line 458. I added the following sentences to understanding the reasons to use docetaxel in NIPS.
Taxans show anti-cancer effects to injure the function of spindle body by promoting microtubule polymerization. Because taxol and docetaxel have hydrophobic property, these drugs are conjugated with cremophol EL and polysorbate 80 to be water soluble (86). Taxol and docetaxel are gradually released from the carriers into ascites after intraperitoneal administration. Accordingly, these drugs tend to stay for a long time after intraperitoneal administration. The concentrations of these drugs in ascites maintains for 24 hours at significantly high levels to kill cancer cells (86).